# Novel pyrimidine-substituted chalcones: In vitro antioxidant properties and cytotoxic effects against human cancer cell lines

Prashant Nayak[1], Vikram S. Shenoy[2], Vijay Upadhye[3], Kasim Sakran Abass[4], Omar Awad Alsaidan[5], Mukesh Soni[6,7], Sunil Tulshiram Hajare[8]*

**1** NGSM Institute of Pharmaceutical Sciences, Deralakatte, Mangalore, India, **2** Innoscience Research Sdn Bhd, Subang Jaya, Selangor, Malaysia, **3** Parul Institute of Applied Science (PIAS), Center of Research for Development (CR4D), Parul University, Vadodara, Gujarat, India, **4** Department of Physiology, Biochemistry, and Pharmacology, College of Veterinary Medicine, University of Kirkuk, Kirkuk, Iraq, **5** Department of Pharmaceutics, College of Pharmacy, Jouf University, Sakaka, Saudi Arabia, **6** Division of Research and Development, Lovely Professional University, Phagwara, Punjab, India, **7** Centre for Research Impact & Outcome, Chitkara University Institute of Engineering and Technology, Chitkara University, Rajpura, Punjab, India, **8** College of Natural and Computational Sciences, Dilla University, Dilla, Ethiopia

* sunilhajare@gmail.com

## Abstract

A series of novel pyrimidine-substituted chalcones were synthesized, purified, and characterized. Four selected compounds (**CFMPY 4, 15, 17, and 28**) exhibited modest antioxidant activity across six *In vitro* radical scavenging assays. The *In vitro* cytotoxic (anticancer) potential of five synthesized chalcones (**CFMPY-2, 4, 15, 17, and 28**) was evaluated using MTT, Sulforhodamine B (SRB), DNA fragmentation, Nuclear staining, and Farnesyl transferase assays. Notably, this study is the first to employ a DNA fragmentation assay to assess the anticancer activity of these test compounds. MTT and SRB assays revealed significant cytotoxicity in HeLa and A549 cell lines for all compounds except **CFMPY 17**, with IC50 values ranging from 2.28 to 5.48 µg/ml, demonstrating comparable or superior efficacy to cisplatin (IC50 values of 5.27 µg/ml in HeLa and 4.05 µg/ml in A549 cells). Nuclear staining and DNA fragmentation assays confirmed the induction of apoptosis by all tested compounds, including cisplatin, with the latter revealing characteristic 200 bp DNA laddering. Furthermore, the Farnesyl transferase assay indicated good cytotoxic activity for all compounds except **CFMPY 17**. These findings suggest that pyrimidine-substituted chalcones represent a promising class of cytotoxic (anticancer) agents, potentially exceeding the efficacy of cisplatin in certain contexts.

## Introduction

Chalcones (1,3-diphenyl-2-propen-1-ones) are linear flavonoids that can be found widely in nature, particularly in fruits, vegetables, teas, and spices. They are

**Data availability statement:** All relevant data are included within the manuscript and its Supporting information files. The original uncropped DNA fragmentation gel images (Figure 5) and raw datasets (antioxidant assay values, cytotoxicity results, and farnesyl transferase assay readings) have been deposited in Figshare and are publicly available at https://doi.org/10.6084/m9.figshare.30030250. Requests for additional data access may be directed to Dr. M. R. Yadhav, Director, RDC, Parul University, Vadodara, Gujarat, India (email: mangeram.yadhav@gmail.com), who serves as a non-author institutional point of contact to ensure long-term accessibility.

**Funding:** The author(s) received no specific funding for this work.

**Competing interests:** The authors have declared that no competing interests exist.

recognized as important precursors in the biosynthetic flavonoid pathway, and their varied structures make them appealing frameworks in medicinal chemistry [1]. In the past five years, there has been a resurgence of interest in chalcones as multifunctional bioactive compounds exhibiting antioxidant, antimicrobial, anti-inflammatory, and anticancer activities [2–4]. Their ability to counteract oxidative stress-related diseases is primarily facilitated through their antioxidant properties, which involve radical scavenging and redox modulation [5]. Additionally, derivatives of chalcones have showcased selective cytotoxic effects against various cancer cell lines, including those from breast, cervical, and colon cancers, frequently by inducing caspase-dependent apoptosis and disrupting mitochondrial pathways [6,7]. These results highlight the potential of chalcones as promising candidates for new drug development [8].

Chalcones have garnered significant interest in pharmacological investigations owing to their broad spectrum of bioactivities. Notably, they exhibit anti-inflammatory properties, a critical factor in the pathogenesis of numerous diseases. Furthermore, chalcones demonstrate antimicrobial activity against a diverse array of microorganisms, suggesting potential applications in the treatment of infectious diseases. Their capacity to function as radical scavengers contributes to the mitigation of oxidative stress and associated pathologies. Parallel to chalcones, pyrimidines have emerged as a privileged scaffold in oncology drug discovery. Reviews from 2020 onward consistently report the diverse pharmacological activities of pyrimidine derivatives, including anticancer, antimicrobial, and immuno-oncological effects [9,10]. In particular, pyrimidine–sulfonamide hybrids and fused pyrimidine analogues show strong multi-targeted anticancer activity by engaging kinase pathways and disrupting oncogenic signaling [11,12]. Importantly, pyrimidine substitution often improves aqueous solubility, hydrogen-bonding capacity, and cellular uptake—features highly desirable for rational anticancer drug design. These multifaceted pharmacological attributes position chalcones as promising lead compounds for the development of novel therapeutic modalities. Contemporary drug design strategies have focused on structural modifications of the chalcone scaffold to enhance their pharmacokinetic profiles, particularly bioavailability, and to elucidate the structure-activity relationships (SAR) associated with various substituents on the aryl or heteroaryl moieties in the context of anticancer drug development. Ongoing research is dedicated to elucidating the underlying molecular mechanisms mediating these biological effects and comprehensively evaluating the therapeutic potential of chalcones.

Farnesyltransferase inhibitors (FTIs) represent a class of experimental anticancer therapeutics that target farnesyltransferase (FTase), an enzyme critical for the post-translational modification of proteins, including the Ras GTPases, which are frequently oncogenically activated in various malignancies [13,14].

Protein prenylation, the post-translational attachment of C15-farnesyl or C20-geranylgeranyl isoprenoid moieties, is catalyzed by a family of transferases. In the context of cancer therapeutics, the inhibition of two key prenyltransferases, FTase and geranylgeranyltransferase-I (GGTase-I), has emerged as a strategy due to their capacity to act on Ras isoforms [15]. These two enzymes are heterodimeric proteins,

sharing a common α-subunit but possessing distinct β-subunits, conferring substrate specificity. Physiologically, in mammalian cells, the post-translational processing of canonical Ras proteins primarily involves FTase-mediated farnesylation. However, upon pharmacological inhibition of FTase by FTIs, proteins such as RhoB, K-Ras, and N-Ras can undergo alternative prenylation by GGTase-I, highlighting a compensatory resistance mechanism.

The substrate recognition site for both FTase and GGTase-I is the conserved carboxyl-terminal tetrapeptide motif, the CAAX box (Cys-Aliphatic-Aliphatic-X), where the terminal amino acid (X) dictates the preference for farnesylation versus geranylgeranylation. Notably, Boguski and McCormick [16] demonstrated that FTase exhibits preferential activity towards protein substrates bearing CAAX motifs terminating in serine, methionine, or glycine. The *In vitro* enzymatic affinity for substrates where X is methionine (e.g., K-Ras A and B with CVIM and CIIM sequences, respectively) is significantly (10–30 fold) higher compared to those terminating in serine or glycine (e.g., H-Ras with CVLS). Conversely, GGTase-I displays preferential activity towards proteins containing leucine as the terminal residue in their CAAX motif.

While all three major Ras isoforms (H-Ras, N-Ras, and K-Ras) are typically farnesylated, the presence of an FTI can lead to K-Ras and N-Ras serving as alternative substrates for GGTase-I. Consequently, K-Ras exhibits increased resistance to FTIs due to this alternative prenylation by GGTase-I, coupled with its inherently higher affinity for FTase. Therefore, a combinatorial therapeutic approach involving both FTIs and geranylgeranyltransferase inhibitors (GGTIs) is posited as necessary to achieve complete blockade of K-Ras prenylation [17], thereby potentially enhancing the efficacy of Ras-targeted cancer therapies.

The post-translational modification of the Ras protein, a key regulator of cell growth and differentiation, is initiated by the farnesylation of its carboxyl-terminal CAAX motif by farnesyltransferase (FTase). This crucial enzymatic step, involving the transfer of a farnesyl moiety from farnesyl pyrophosphate (FPP), is essential for the proper localization and function of Ras at the cell membrane [18]. Aberrant activation of Ras isoforms is a frequent occurrence in a wide range of human cancers, driving uncontrolled cell proliferation, invasiveness, and metastatic potential – the defining hallmarks that distinguish malignant from benign tumors [19]. Given that virtually any living cell in the body can undergo malignant transformation, and considering the diverse etiologies of various cancer types, the development of effective and targeted anticancer therapies remains a paramount challenge in biomedical research. In this context, the exploration of novel therapeutic strategies that disrupt oncogenic Ras signaling pathways holds immense significance. Farnesyltransferase inhibitors (FTIs) have emerged as a promising class of anticancer agents by specifically blocking the initial farnesylation step of Ras, thereby impairing its downstream signaling and potentially inhibiting tumor growth. Despite extensive reports on the diverse bioactivities of chalcones, their therapeutic utility is often limited by modest potency and non-specificity. Rational structural modification of the chalcone scaffold has been shown to improve pharmacological profiles, particularly in the context of anticancer activity. Among various heterocyclic substitutions, the pyrimidine moiety is a privileged scaffold in medicinal chemistry due to its well-documented ability to enhance bioavailability, strengthen hydrogen-bonding interactions with biological targets, and confer improved cytotoxic selectivity. Furthermore, pyrimidine-based compounds have previously demonstrated activity against oncogenic signaling pathways, including Ras-dependent cascades, suggesting a potential for farnesyltransferase inhibition. To emphasize the structural novelty of our design, a schematic comparison (Fig 1) is provided, illustrating similarities and differences between the synthesized pyrimidine-substituted chalcones, previously reported chalcones with related antioxidant/cytotoxic activity, and representative pyrimidine-based drugs.

Therefore, we hypothesized that pyrimidine substitution in chalcones could synergize radical-scavenging with Ras-targeted anticancer activity. To test this, we synthesized and evaluated pyrimidine-substituted chalcones for antioxidant and anticancer potential across multiple cell lines, using assays including MTT, SRB, nuclear staining, DNA fragmentation, and FTase inhibition.

The present study specifically aims to explore the anticancer potential of pyrimidine-substituted chalcones by evaluating their cytotoxic effects against human cancer cell lines and investigating apoptosis and farnesyltransferase inhibition as possible mechanisms. To achieve this, we employed a multifaceted approach encompassing MTT and Sulforhodamine B assays, nuclear staining, DNA fragmentation, and a farnesyl transferase assay.

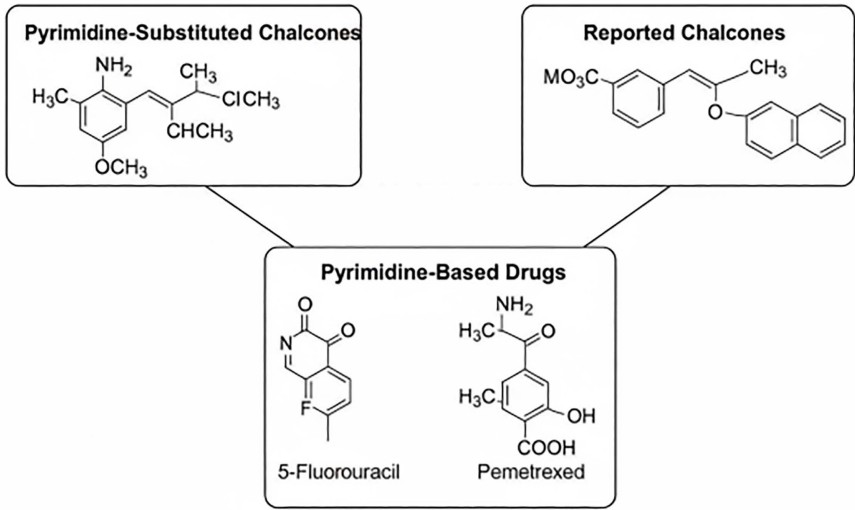

**Fig 1. Schematic comparison illustrating the structural novelty of the synthesized pyrimidine-substituted chalcones.** The structures of the synthesized pyrimidine-substituted chalcones are compared with previously reported chalcones and known pyrimidine-based anticancer drugs such as 5-Fluorouracil and Pemetrexed.

## Materials and methods

### Experimental synthesis of test compounds

Fig 2 outlines the synthetic route for pyrimidine-substituted chalcone derivatives. Chalcones were synthesized *via* a Claisen-Schmidt condensation. Briefly, 2-fluoro-4-methoxy acetophenone (122.5 mL) was reacted with various substituted benzaldehydes in a solution of sodium hydroxide (22.5 g in 200 mL water, corresponding to ~2.8 M NaOH) and rectified spirit (122.5 mL) and rectified spirit (122.5 mL) under cold conditions (ice bath) with mechanical stirring. The reaction mixture was stirred overnight at refrigerated temperatures. The resulting crude chalcones were then isolated by filtration and purified by recrystallization from ethanol [20,21].

Guanidine hydrochloride and the previously synthesized chalcone derivatives were dissolved in absolute ethanol. A solution of potassium hydroxide (prepared by dissolving the required amount in a minimal volume of distilled water, specifically twice the mass of the chalcones and guanidine hydrochloride combined) was added to the ethanolic reaction mixture. The resulting mixture was then refluxed for 6 hours. Following the reflux period, the reaction mixture was poured into 250 mL of ice-cold distilled water to induce precipitation and facilitate crystallization of the pyrimidine-substituted chalcones. The resulting solid was then typically collected by filtration and further purified (e.g., by recrystallization) to obtain the desired compounds [22].

Compound purity was assessed using ascending Thin Layer Chromatography (TLC) on pre-coated Silica gel-G plates, with visualization of spots achieved by exposure to iodine vapors. A computational docking study using the Schrödinger software package was employed to select compounds **CFMPY2, CFMPY4, CFMPY15, CFMPY17**, and **CFMPY28** for further biological evaluation. Structural characterization of the synthesized compounds was performed using spectroscopic and analytical techniques, including Infrared (IR) spectroscopy, Proton Nuclear Magnetic Resonance ($^1$H NMR) spectroscopy, mass spectrometry, and elemental analysis. IR spectra were recorded as potassium bromide (KBr) discs on a Shimadzu IR Affinity FT-IR spectrophotometer, and characteristic absorption bands are reported in wave numbers (cm$^{-1}$). $^1$H NMR spectra were acquired on a Bruker AC 400 MHz spectrometer using Deuterated dimethyl sulfoxide (DMSO-D6) as the solvent, with tetramethylsilane (TMS) serving as the internal chemical shift reference. Chemical

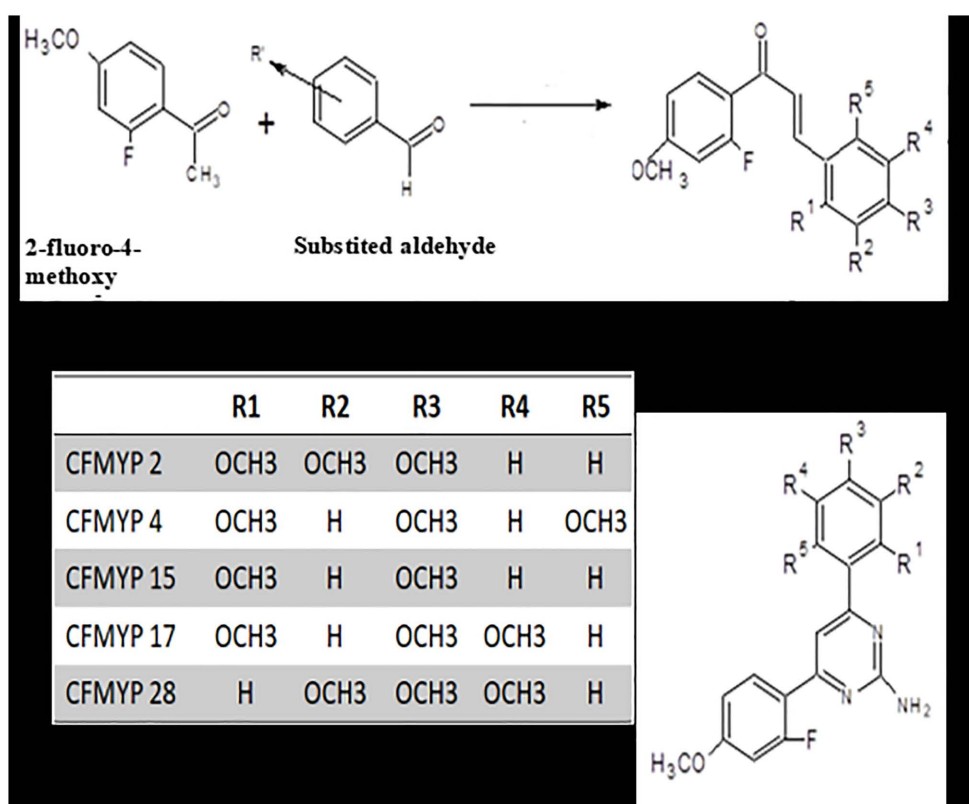

**Fig 2. Synthetic scheme for the preparation of 2-amino-4-(2-fluoro-4-methoxyphenyl)-6-substituted phenylpyrimidine derivatives (CFMPY-2, 4, 15, 17, and 28).** The scheme illustrates the Claisen–Schmidt condensation followed by cyclization with guanidine hydrochloride.

shifts (δ) are reported in parts per million (ppm). Mass spectral data and elemental analysis results were used to further confirm the molecular structures and purity of the synthesized compounds

**2-amino-4-(2-fluoro-4-methoxyphenyl)-6-(2,3,4-trimethoxyphenyl)pyrimidine (CFMPY-2).** IR KBr (cm⁻¹): 3056 (Ar C-H str), 1580 (Ar C=C str), 830 (Ar C-H bend), 1636 (C=N str), 3380 (N-H str), 1205 (C-F str), 1226 (C-O str). ¹H NMR (400 MHz, CDCl₃, δ ppm): 3.61 (3H, s), 3.83 (3H, s), 3.87 (3H, s), 3.84 (3H, s), 6.43 (1H, d, J=8.4 Hz), 7.10–7.11 (2H, 7.10 (s), 7.01 (dd, J=8.4, 1.5 Hz), 7.20 (1H, dd, J=1.5, 0.5 Hz), 7.54–7.62 (2H, 7.58 (d, J=8.5 Hz), 7.63 (dd, J=8.4, 0.5 Hz). MS (M⁺): 386.

**2-amino-4-(2-fluoro-4-methoxyphenyl)-6-(2,4,6-trimethoxyphenyl)pyrimidine (CFMPY-4).** IR KBr (cm⁻¹): 3058 (Ar C-H str), 1582 (Ar C=C str), 836 (Ar C-H bend), 1630 (C=N str), 3397 (N-H str), 1202 (C-F str), 1223 (C-O str). ¹H NMR (400 MHz, CDCl₃, δ ppm): 3.82–3.84 (6H, 3.84 (s), 3.83 (s)), 3.86 (6H, s), 6.35 (2H, d, J=1.8 Hz), 7.10 (1H, s), 7.12 (1H, dd, J=8.8, 1.6 Hz), 7.31 (1H, dd, J=1.6, 0.5 Hz), 7.58 (1H, dd, J=8.8, 0.5 Hz). MS (M⁺): 386.

**2-amino-4-(2-fluoro-4-methoxyphenyl)-6-(2,4-dimethoxyphenyl)pyrimidine(CFMPY-15).** IR KBr (cm⁻¹): 3050 (Ar C-H str), 1586 (Ar C=C str), 828 (Ar C-H bend), 1622 (C=N str), 3387 (N-H str), 1206 (C-F str), 1230 (C-O str). ¹H NMR (400 MHz, CDCl₃, δ ppm): 3.83–3.84 (6H, 3.84 (s), 3.84 (s)), 3.89 (3H, s), 6.56 (1H, dd, J=1.6, 0.5 Hz), 6.97 (1H, dd, J=8.6, 1.6 Hz), 7.09–7.15 (2H, 7.10 (s), 7.13 (dd, J=8.6, 1.6 Hz)), 7.28 (1H, dd, J=1.6, 0.5 Hz), 7.63 (1H, dd, J=8.6, 0.5 Hz), 7.72 (1H, dd, J=8.6, 0.5 Hz). MS (M⁺): 355.

**2-amino-4-(2-fluoro-4-methoxyphenyl)-6-(2,4,5-trimethoxyphenyl)pyrimidine (CFMPY-17).** IR KBr (cm⁻¹): 3052 (Ar C-H str), 1579 (Ar C=C str), 832 (Ar C-H bend), 1618 (C=N str), 3385 (N-H str), 1208 (C-F str), 1232 (C-O str). ¹H

The scheme table:

|  | R1 | R2 | R3 | R4 | R5 |
|---|---|---|---|---|---|
| CFMYP 2 | OCH3 | OCH3 | OCH3 | H | H |
| CFMYP 4 | OCH3 | H | OCH3 | H | OCH3 |
| CFMYP 15 | OCH3 | H | OCH3 | H | H |
| CFMYP 17 | OCH3 | H | OCH3 | OCH3 | H |
| CFMYP 28 | H | OCH3 | OCH3 | OCH3 | H |

NMR (400 MHz, CDCl$_3$, δ ppm): 3.61 (3H, s), 3.82 (3H, s), 3.83 (3H, s), 3.82 (3H, s), 6.52 (1H, d, J = 0.4 Hz), 7.05–7.10 (2H, 7.10 (s), 7.09 (dd, J = 8.4, 1.5 Hz)), 7.18 (1H, dd, J = 1.5, 0.5 Hz), 7.42 (1H, d, J = 0.4 Hz), 7.61 (1H, dd, J = 8.4, 0.5 Hz). MS (M$^+$): 386.

**2-amino-4-(2-fluoro-4-methoxyphenyl)-6-(3,4,5-trimethoxyphenyl)pyrimidine (CFMPY-28).** IR KBr (cm$^{-1}$): 3048 (Ar C-H str), 1578 (Ar C = C str), 826 (Ar C-H bend), 1625 (C = N str), 3382 (N-H str), 1206 (C-F str), 1235 (C-O str). $^1$H NMR (400 MHz, CDCl$_3$, δ ppm): 3.64 (6H, s), 3.83 (3H, s), 3.81 (3H, s), 7.09 (1H, dd, J = 8.4, 1.5 Hz), 7.13 (1H, dd, J = 1.5, 0.5 Hz), 7.20 (1H, s), 7.26 (2H, d, J = 2.4 Hz), 7.63 (1H, dd, J = 8.4, 0.5 Hz). MS (M$^+$): 386.

## MTT assay

Adherent cell lines were grown in Dulbecco's Modified Eagle's Medium (DMEM), enriched with 10% (v/v) heat-inactivated Fetal Bovine Serum (FBS; please indicate the supplier and batch number if it is crucial). The conditions were maintained at 37°C in a humid atmosphere with 5% $CO_2$. For the assessment of *In vitro* cytotoxicity, cells were plated in 96-well microplates (such as those made of tissue culture-treated polystyrene) at a specific seeding density (cells/well) to guarantee sub-confluent growth throughout the assay duration and allowed to adhere for a duration of 24 hours. After the cells had adhered, test compounds were added in serial dilutions (usually $log_{10}$ or two-fold) in the culture medium. Control wells received a corresponding concentration of the solvent (for example, DMSO, with a maximum final concentration of ≤ 0.1% v/v) to mitigate potential effects from the solvent. The cells were then incubated for 72 hours under standard culture conditions. After incubation, the culture medium was removed, and replaced with a sterile-filtered MTT solution (for instance, 0.5 mg/mL in sterile PBS). Following a 2–4 hour incubation at 37°C, the MTT solution was carefully discarded, and the intracellular formazan crystals were dissolved by adding a specific volume of an appropriate solvent (such as anhydrous DMSO or isopropanol containing 0.04 N HCl). The absorbance was measured using a microplate spectrophotometer (ELISA reader) at a primary wavelength of 570 nm and a reference wavelength of 630 nm to reduce background interference. Cytotoxicity was determined as a percentage in relation to the mean absorbance of the untreated control wells utilizing the following formula:

$$\text{\% Cytotoxicity} = (\text{Absorbance of control} - \text{Absorbance of test}) / (\text{Absorbance of control}) \times 100$$

## Sulphorhodamine B assay

Cells were cultured in Dulbecco's modified essential medium with 10% FBS and fetal bovine serum. The cell suspension was diluted and applied to 96-well microliter plates. The plates were incubated at 37°C for 24 hours, then drug concentrations were applied. After 72 hours, a 10% concentration of trichloroacetic acid was added. The plates were cleaned, air dried, and then dissolved in a sample containing 10mM tris base. The absorbance of each well was measured at 540nm using an ELISA reader, and the percentage cytotoxicity was calculated.:

$$\text{\% Cytotoxicity} = (\text{Absorbance of control} - \text{Absorbance of test}) / (\text{Absorbance of control}) \times 100$$

## DNA fragmentation assay

*Cell harvesting.* HeLa cells (5 x 10$^5$ - 1 x 10$^6$) were harvested and pelleted by centrifugation at 4000 rpm (approximately 2500 x g) for 5 minutes at 4°C in a 1.5mL microcentrifuge tube. The supernatant was carefully aspirated and retained for potential downstream analyses (e.g., protein assays), flash-frozen in liquid nitrogen, and stored at −20°C. A brief secondary centrifugation (15 seconds at the same conditions) was performed to ensure complete removal of the supernatant.

**Cell lysis and protein digestion.** The cell pellet was resuspended by vigorous vortexing in 20 µL of Lysis Buffer I (composition: EDTA to chelate divalent cations inhibiting nuclease activity, Proteinase K for protein degradation including histones and DNases, and Sodium Lauryl Sarcosinate (SLS) to denature proteins and release DNA). The resulting lysate was incubated at 50°C for 1 hour in a temperature-controlled heating block or water bath.

**RNA digestion.** Following incubation, any condensate on the microcentrifuge tube lid was collected by brief centrifugation. The lysate was then treated with 10 µL of RNase A solution (concentration specified, e.g., 10 mg/mL, DNase-free), vortexed thoroughly, and incubated at 50°C for 1 hour to eliminate RNA contamination.

**Agarose gel preparation.** A 2% (w/v) agarose gel was prepared by dissolving agarose in Tris-Phosphate-EDTA (TPE) buffer (composition specified, e.g., 89 mM Tris base, 89 mM phosphoric acid, 2 mM EDTA, pH 8.0) using microwave heating. After cooling the molten agarose to approximately 50–55°C, ethidium bromide (EtBr) was added to a final concentration of x µg/mL (e.g., 0.5 µg/mL) for DNA visualization. The agarose-EtBr solution was immediately poured into a gel casting tray with a well-forming comb positioned approximately 1 cm from one end, ensuring the gel solidified on a level surface for at least 20 minutes at room temperature.

**DNA sample loading and electrophoresis.** After RNA digestion, the cell lysate was briefly centrifuged to collect any condensation. 10 µL of 6X DNA loading buffer (composition specified, e.g., containing a tracking dye and a density agent) that had been pre-heated to 70°C was added to the DNA sample. The mixture was briefly vortexed and centrifuged. The well-forming comb was carefully removed from the solidified agarose gel, and the gel was submerged in TPE buffer in the electrophoresis tank. The prepared DNA samples were carefully loaded into the wells. Electrophoresis was performed at a constant voltage of 50 V for a defined duration (e.g., 30 minutes) to allow for adequate separation of DNA fragments based on size.

**DNA visualization and documentation.** Following electrophoresis, the gel was visualized under a UV transilluminator of a gel documentation system. The presence of a characteristic ladder-like pattern of DNA fragments, representing multiples of approximately 200 base pairs, indicates internucleosomal DNA fragmentation, a hallmark of apoptosis. Images of the gel were captured using the gel documentation system for analysis and record-keeping. A DNA size standard (ladder) should be run in parallel to determine the size of the DNA fragments [23].

## Nuclear staining method

HeLa cells were seeded in a 6-well plate, incubated at 37°C, washed, centrifuged, and resuspended in a solution containing acridine orange and ethidium bromide in PBS. The cells were then visualized using a fluorescence microscope using a blue filter, and the supernatant was discarded after the incubation period [24].

## Farnesyl transferase assay

**Preparation of cytosolic fractions.** Adult male Wistar rats (obtained from Manipal Life Sciences) were humanely euthanized using chloroform inhalation, adhering to ethical guidelines. The brain was quickly extracted and placed in ice-cold 0.1 M HEPES buffer (pH 7.4) containing 25 mM MgCl$_2$ and 10 mM DTT. The tissue was meticulously cleared of meninges and blood vessels, finely diced, and homogenized in the same ice-cold buffer using a Polytron homogenizer (ten bursts at 800 rpm with intermittent cooling). The homogenate was centrifuged at 10,000 rpm (specific rotor and g-force should be indicated for reproducibility) for 30 minutes at 4°C. The supernatant obtained was then subjected to ultracentrifugation at 100,000 rpm (specific rotor and g-force should be detailed) for 60 minutes at 4°C to isolate the cytosolic fraction (supernatant), which was then aliquoted into 1 mL cryovials and preserved at −80°C. The protein concentration in the cytosolic fraction was determined using the Hartree-Lowry method, with bovine serum albumin (BSA) serving as a standard [25].

The Farnesyltransferase (FTase) activity assay was conducted in a 96-well microplate format. Each reaction mixture contained 30 µg of rat brain cytosolic protein (prepared as described previously), a defined concentration of the

fluorescent dansyl peptide substrate (ranging from 1.6 to 6 µM to assess substrate dependence), and a detergent (concentration and type specified, e.g., 0.05% Triton X-100) in a total volume of V µL (calculated based on component volumes) of assay buffer. The assay buffer consisted of 52 mM Tris-HCl (pH 7.4), supplemented with 5.0 mM DTT, 12 mM MgCl$_2$, and 12 µM ZnCl$_2$. The reaction mixtures were pre-incubated at a stable temperature of 30°C for 2 minutes in a temperature-controlled microplate reader to ensure thermal equilibration. The enzymatic reaction was initiated by the automated addition of 25 µM farnesyl pyrophosphate (FPP). The contents of each well were rapidly and uniformly mixed by orbital shaking within the microplate reader.

The kinetics of the farnesylation reaction were monitored by continuously recording the fluorescence intensity at a fixed excitation wavelength of 340 nm and an emission wavelength of 530 nm using a fluorescence microplate reader (model specified). Fluorescence measurements were acquired over a period of 750 seconds, and the initial linear rate of the reaction, expressed as counts per second per second (cps/s), was determined by linear regression analysis of the fluorescence intensity versus time plot during the initial phase of the reaction where substrate depletion is negligible. All measurements for each condition were performed in triplicate (n = 3), and the mean and standard deviation (SD) or standard error of the mean (SEM) were calculated.

Control experiments were performed to assess the endogenous fluorescence contribution from the rat brain cytosol. In the absence of the dansyl peptide substrate, the basal increase in fluorescence intensity at 530 nm, attributed to endogenous fluorophores within the cytosol, was observed to be approximately 10% of the initial fluorescence value and remained stable over the assay duration, indicating minimal interference with the measurement of the enzymatic reaction. This background fluorescence was accounted for in subsequent data analysis. Full characterization data, including IR, ^1H and ^13C NMR spectra (with integration and chemical structures), and MS/HRMS spectra for all final compounds (CFMPY-2, CFMPY-4, CFMPY-15, CFMPY-17, and CFMPY-28), are provided in the Supporting Information (S1–S16 Figs).

## Results

The cytotoxic activity of each compound was evaluated using the MTT assay. HeLa, A549, and HepG2 cell lines were exposed to the compounds at a concentration range of 100 to 1.5625 µg/ml. Results indicated that all tested compounds exhibited cytotoxic effects against all three cell lines (Fig 3). As depicted in Fig 3, the compounds demonstrated a more significant inhibitory effect on HeLa and A549 cell proliferation compared to HepG2 cells, where the activity was comparable to that of the reference drug cisplatin. Consequently, the Sulforhodamine B (SRB) assay was employed to further assess the compounds' growth inhibitory effects specifically against HeLa and A549 cell lines.

### SRB assay

The synthesized compounds were evaluated for their cytotoxic potential against HeLa and A549 cell lines. The results revealed a range of cytotoxic activities among the tested compounds. Specifically, compound CFMPY-2 exhibited the most potent cytotoxicity in HeLa cells, while CFMPY 4 demonstrated the highest cytotoxicity in A549 cells. Conversely, compound CFMPY 15 displayed the lowest cytotoxic effect in HeLa cells, and CFMPY 17 showed the lowest cytotoxicity in A549 cells (Fig 4). Notably, several compounds demonstrated superior cytotoxicity compared to the reference standard cisplatin in both HeLa and A549 cell lines, suggesting their potential as promising therapeutic agents.

### DNA fragmentation assay

Apoptosis was assessed using the DNA fragmentation assay, which detects the characteristic ladder-like pattern of DNA fragments resulting from internucleosomal cleavage. The presence of bands at approximately 200 base pairs (bps) and

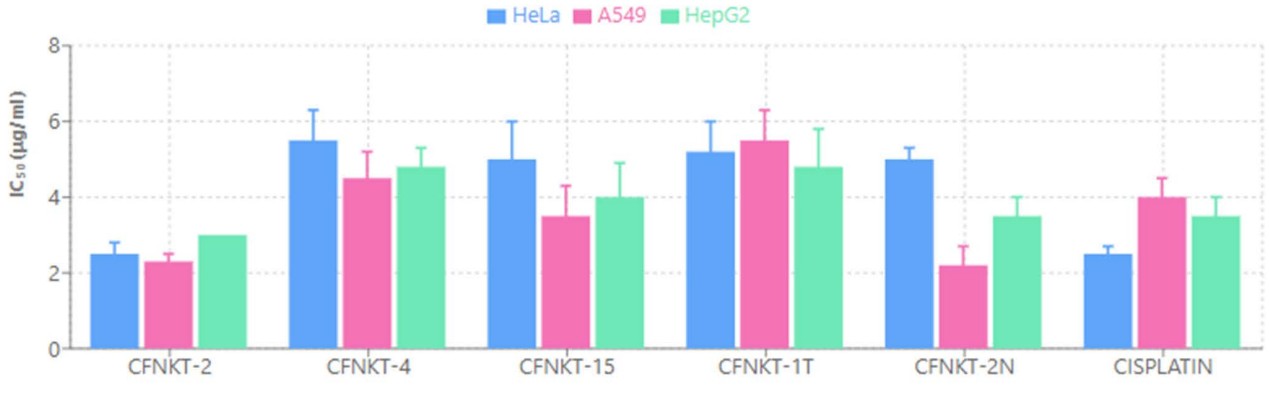

**Fig 3. Cytotoxic effects of the synthesized compounds against HeLa, A549, and HepG2 cell lines using the MTT assay.** Results show percent viability after 72 hours of treatment with increasing concentrations.

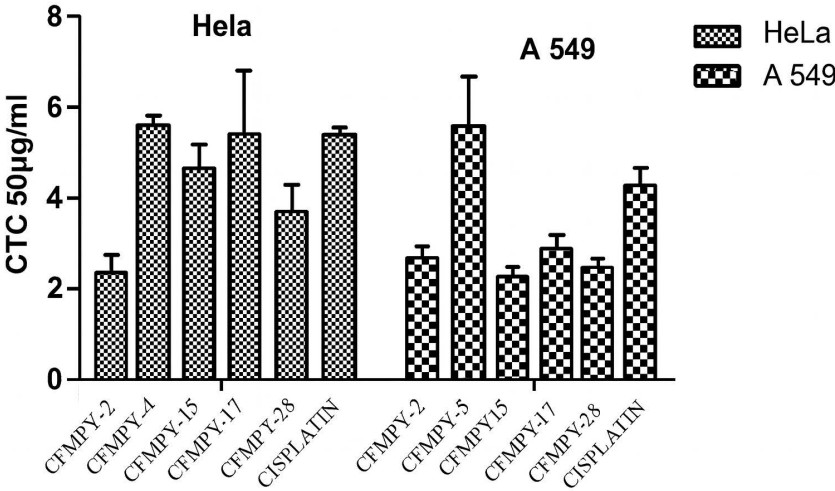

**Fig 4. Cytotoxic activity of the synthesized compounds against HeLa and A549 cell lines using the Sulforhodamine B (SRB) assay.** Compounds CFMPY-2 and CFMPY-4 exhibited the highest cytotoxicity.

multiples thereof is indicative of apoptotic DNA fragmentation. As depicted in Fig 5 (uncropped image in S17 Fig), no such ladder-like banding pattern was observed, no such ladder-like banding pattern was observed in cells treated with any of the tested compounds or cisplatin. However, further investigation using nuclear staining techniques revealed significant apoptotic activity in cells treated with both cisplatin and all tested compounds. These findings suggest that while the compounds and cisplatin induce apoptosis, the mechanism may not primarily involve the classical internucleosomal DNA fragmentation detectable by the ladder assay under the experimental conditions employed. Therefore, the tested substances possess the ability to induce apoptosis through pathways potentially distinct from or upstream of extensive DNA fragmentation into the typical ladder pattern.

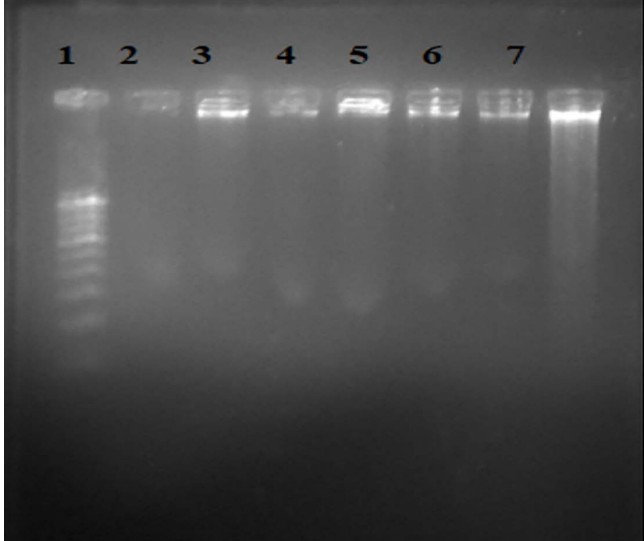

**Fig 5. DNA fragmentation assay of HeLa cells treated with synthesized compounds.** The absence of classical DNA laddering suggests non-internucleosomal fragmentation under the test conditions. Lane assignments: 1 – DNA Ladder, 2 - CFMPY-2, 3 - CFMPY-4, 4 - CFMPY-15, 5 - CFMPY-17, 6 - CFMPY-28, 7 – Cisplatin. The uncropped gel image is available in S17 Fig.

## Nuclear staining

Acridine orange/ethidium bromide staining of HeLa cells revealed distinct nuclear changes (Fig 6). Late apoptotic cells appeared orange with chromatin condensation and clumping (blue arrows), while early apoptotic cells appeared green with highly condensed chromatin (white arrows). Following compound treatment, only a few viable green cells were observed; most cells appeared orange without chromatin condensation, indicating necrotic rather than apoptotic death.

## Farnesyl transferase assay

With the exception of the CFMPY 17, every compound in the Farnesyl transferase test as displayed in Fig 7 had strong cytotoxic activity. Prior to doing more research on FTase inhibitors, the process has to be standardized because the Ic 50 value of these inhibitors was reported to be in nanomoles (Table 1).

## Structure–activity relationship

Following the evaluation of both antioxidant and cytotoxic activities, we explored whether a relationship exists between the two. Interestingly, our analysis revealed that higher antioxidant potential did not consistently correlate with stronger cytotoxic effects. Nonetheless, certain pyrimidine-substituted chalcones exhibited overlapping profiles, suggesting that specific substitution patterns may contribute to both activities. These findings provide a useful context for interpreting the structure–activity relationships (SAR) and highlight possible mechanistic links between antioxidant defense and cytotoxic response [26; Fig 8].

## Discussion

The pyrimidine-substituted chalcones were synthesized through a modified Claisen–Schmidt condensation, a standard and well-established route for chalcone derivatives [22]. In this method, substituted benzaldehydes were condensed with 2-fluoro-4-methoxyacetophenone under ethanolic KOH to yield α,β-unsaturated ketone intermediates, which were subsequently cyclized with guanidine hydrochloride to afford the desired pyrimidine derivatives [22,23]. The synthesized

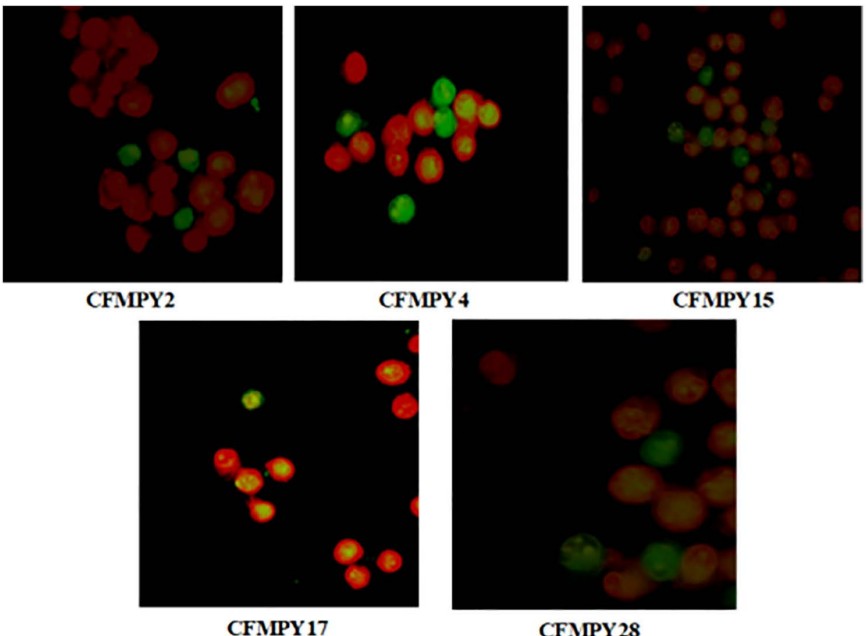

**Fig 6. Morphological changes observed in HeLa cells after treatment with synthesized compounds using acridine orange/ethidium bromide nuclear staining.** Apoptotic and necrotic features are visualized under fluorescence microscopy.

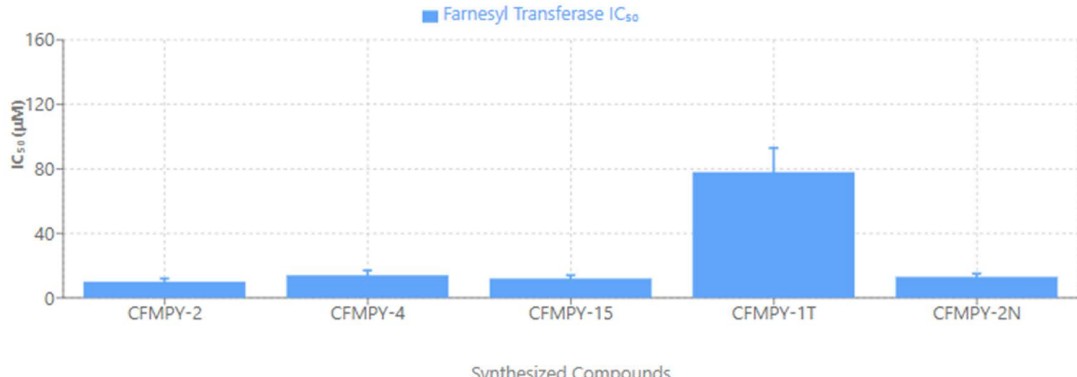

**Fig 7. Farnesyltransferase (FTase) inhibition assay showing cytotoxic activity of synthesized compounds.** All compounds except CFMPY-17 exhibited significant FTase inhibitory potential.

compounds were purified by recrystallization, and their purity confirmed by thin-layer chromatography (TLC), a reliable analytical tool for monitoring compound separation. This robust synthetic approach enabled the preparation of a structurally diverse library of pyrimidine-based chalcone derivatives for biological evaluation.

To investigate their anticancer potential, a series of **in vitro assays** was employed, in alignment with established methodologies for early drug discovery [26–30]. Cytotoxicity was initially assessed using the MTT assay, which quantifies mitochondrial reduction of MTT to formazan as an indicator of cellular metabolic activity [27]. Complementary evaluation was conducted via the Sulforhodamine B (SRB) assay, which measures protein content as a surrogate for cell proliferation.

**Table 1. Comparisons of IC50s of the test compounds in the various radical scavenging assays.**

| Test Compounds | DPPH-free radical scavenging IC50 (µg/mL) | ABTS radical scavenging IC50 (µg/mL) | Hydrogen peroxide radical scavenging IC50 (µg/mL) | Nitric oxide radical scavenging IC50 (µg/mL) | Superoxide radical scavenging IC50 (µg/mL) | Total antioxidant capacity IC50 (µg/mL) |
|---|---|---|---|---|---|---|
| CFMPY-2 | 733.65 | *78.50* | *8.55* | >1000 | >1000 | >1000 |
| CFMPY 4 | 369.38 | 119.79 | 37.30 | >1000 | >1000 | >1000 |
| CFMPY 15 | >1000 | 258.16 | >1000 | >1000 | >1000 | >1000 |
| CFMPY 17 | >1000 | >1000 | >1000 | >1000 | >1000 | >1000 |
| CFMPY-28 | >1000 | 323.21 | 868.5 | >1000 | >1000 | >1000 |
| Ascorbic acid | 12.12 | 103.39 | 8.97 | 189.20 | 8.86 | 12.17 |

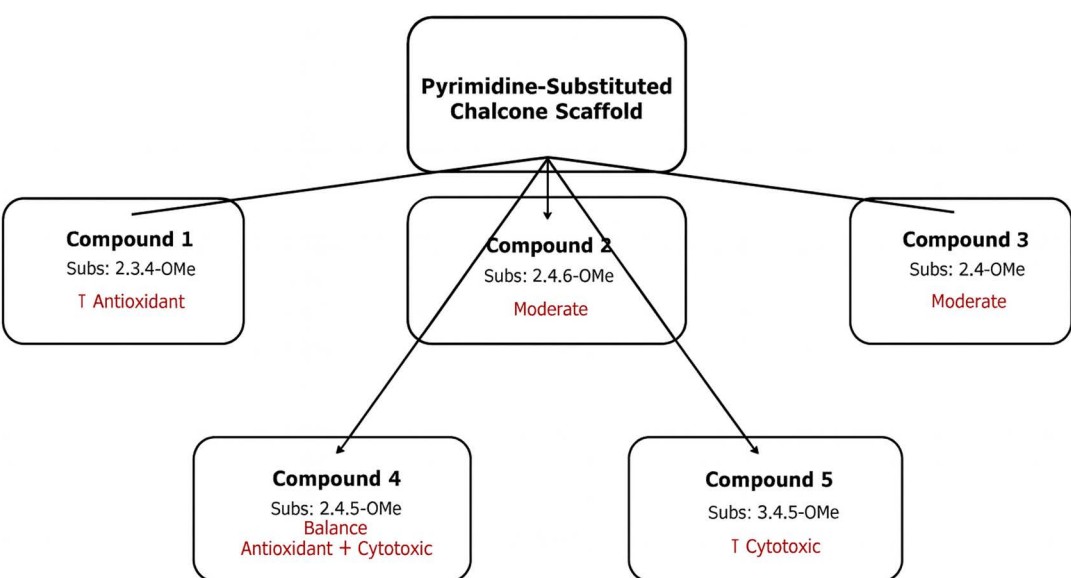

**Fig 8. Structure–Activity Relationships (SAR) of the synthesized pyrimidine-substituted chalcones.** The figure summarizes correlations between antioxidant activity, cytotoxicity, and structural substitutions.

Together, these assays provided a reliable dual perspective on cytotoxic effects, balancing metabolic and structural endpoints.

The results revealed broad-spectrum cytotoxicity across the tested human cancer cell lines—HeLa (cervical carcinoma), A549 (lung carcinoma), and HepG2 (hepatocellular carcinoma) [31–33]. Notably, HeLa and A549 cells exhibited greater sensitivity than HepG2 cells, suggesting possible selectivity toward cervical and lung malignancies. Importantly, the activity against these two cell lines was comparable to cisplatin, a frontline platinum-based chemotherapeutic agent included as a positive control [34,35]. This observation underscores the promise of these novel derivatives, especially given cisplatin's limitations related to toxicity and resistance.

The SRB assay corroborated the MTT results, particularly emphasizing the cytotoxic potency of compounds CFMPY-2 and CFMPY-4 in HeLa and A549 cells, respectively. By contrast, CFMPY-15 and CFMPY-17 displayed weaker effects, likely reflecting reduced affinity for cellular targets or the need for higher concentrations to exert measurable activity [26].

These findings are consistent with the independent study of Ballinas-Indili et al. [36], who also observed high potency of CFMPY-2 and CFMPY-4 in the same cell lines, occasionally surpassing cisplatin. Such convergence across independent investigations strengthens the evidence for these compounds as promising lead candidates. Furthermore, our results align with broader insights from Gharbaran et al. [7], who highlighted the utility of novel cytotoxic scaffolds, and complement alternative approaches such as nanoparticle-mediated synergy [37]. Collectively, the selective efficacy of CFMPY-2 and CFMPY-4 underscores their strong potential for further development.

Mechanistic studies of apoptosis provided deeper insights into the mode of cell death. Nuclear staining with acridine orange and ethidium bromide revealed hallmark apoptotic features, including chromatin condensation and nuclear fragmentation [29,38]. However, the DNA fragmentation assay did not yield the characteristic ladder pattern of internucleosomal cleavage, highlighting a discordance between biochemical and morphological markers. This discrepancy underscores the limitations of relying solely on DNA fragmentation as definitive evidence of apoptosis. Several explanations are plausible: the compounds may activate non-classical apoptotic pathways, apoptosis could occur in a caspase-independent manner, or DNA fragmentation may be transient and missed at the point of analysis [39]. The observation of apoptotic nuclear morphology despite absent DNA laddering is consistent with reports of caspase-independent or delayed fragmentation mechanisms. Together, these findings emphasize the heterogeneity of apoptotic processes and validate the necessity of employing complementary approaches—integrating morphological assays with biochemical methods—for accurate apoptosis assessment.

Beyond cytotoxicity and apoptosis, **farnesyltransferase (FTase) inhibition** was evaluated to explore potential effects on oncogenic signaling. All compounds, with the exception of CFMPY-17, displayed notable FTase inhibitory activity, suggesting disruption of Ras-mediated pathways, which are central to cancer progression [30]. These results highlight the potential dual mechanism of the synthesized compounds: direct cytotoxicity and indirect modulation of oncogenic signaling. Further determination of $IC_{50}$ values is warranted to enable direct comparison with established FTase inhibitors. Previous reports describe nanomolar-potency FTase inhibitors [27,40], setting a benchmark for future optimization. Insights from Carta et al. [41], who identified compound 5f as a potent anticancer agent acting through tubulin polymerization inhibition, cell cycle arrest, caspase-dependent apoptosis, and multi-kinase inhibition, further reinforce the therapeutic relevance of multi-targeted agents. In this context, pyrimidine-substituted chalcones may similarly offer pleiotropic anticancer mechanisms.

An additional layer of therapeutic potential is suggested by the **integration of antioxidant and cytotoxic activities** within these chalcone derivatives. Oxidative stress, a critical driver of DNA damage, genomic instability, and tumorigenesis, remains a central challenge in oncology [42]. Molecules that combine radical-scavenging properties with selective cytotoxicity may simultaneously mitigate oxidative injury in normal tissues while exerting lethal effects on malignant cells, embodying a multifunctional therapeutic paradigm [43–45]. Thus, the observed cytotoxic and possible antioxidant effects of pyrimidine-substituted chalcones represent a valuable strategy for cancers where oxidative stress is a key pathogenic factor.

## Conclusion

The Claisen-Schmidt condensation successfully yielded pyrimidine-substituted chalcones, confirmed and purified through spectral analysis. While *In vitro* radical scavenging assays indicated modest antioxidant activity for compounds CFMPY 4, CFMPY 15, CFMPY 17, and CFMPY-28, *In vitro* cytotoxicity testing revealed promising anticancer potential for CFMPY 4, CFMPY 15, and CFMPY-28, demonstrating good efficacy compared to cisplatin. However, the lack of *In vitro* efficacy for CFMPY 17 underscores the structural activity relationship at play. Crucially, the observed *In vitro* cytotoxic activity does not guarantee *in vivo* success. Therefore, comprehensive *in vivo* studies are now essential to bridge this gap. These studies will be critical in elucidating the pharmacokinetics (absorption, distribution, metabolism, excretion), biodistribution (tissue penetration), and potential toxicity profiles of these compounds in a living system. Ultimately, these *in vivo*

investigations will determine which of the synthesized compounds, particularly CFMPY 4, CFMPY 15, and CFMPY-28, possesses the most favorable balance of efficacy and safety, thus identifying the most promising candidate(s) for further development as a potential anticancer drug.

**Limitations of study**

A limitation of the present study is the absence of cytotoxicity assays in non-tumorigenic (normal) cell lines. Such experiments (e.g., BEAS-2B, Ect1/E6E7) are underway to establish selectivity indices (SI = $IC_{50}$ normal/ $IC_{50}$ cancer) and will further strengthen the therapeutic relevance of our scaffold.

## Supporting information

**S1 Fig.** [1]H NMR spectrum of CFMPY-2. Shows aromatic and methoxy proton regions confirming the structure.
(PDF)

**S2 Fig.** [1]H NMR spectrum of CFMPY-4. Indicates methoxy and aromatic proton patterns supporting substitution.
(PDF)

**S3 Fig.** [1]H NMR spectrum of CFMPY-15. Displays characteristic proton splitting of substituted phenyl ring.
(PDF)

**S4 Fig.** [1]H NMR spectrum of CFMPY-17. Shows defined peaks consistent with methoxy-substituted pyrimidine.
(PDF)

**S5 Fig.** [1]H NMR spectrum of CFMPY-28. Depicts proton signals for methoxy and aromatic regions.
(PDF)

**S6 Fig.** Enlarged spectral view for CFMPY-2. Highlights specific downfield shifts corresponding to electron-rich substituents.
(PDF)

**S7 Fig.** Enlarged spectral view for CFMPY-4. Focuses on methoxy group shifts and splitting pattern.
(PDF)

**S8 Fig.** Enlarged spectral view for CFMPY-15. Emphasizes coupling constants and ring proton patterns.
(PDF)

**S9 Fig.** Enlarged spectral view for CFMPY-17. Clarifies overlapping regions in the aromatic range.
(PDF)

**S10 Fig.** Enlarged spectral view for CFMPY-28. Displays proton environments around pyrimidine core.
(PDF)

**S11 Fig.** Secondary NMR region for CFMPY-2. Shows cleaner resolution of overlapping signals.
(PDF)

**S12 Fig.** Secondary NMR region for CFMPY-4. Further confirms signal integrity for aromatic protons.
(PDF)

**S13 Fig.** Secondary NMR region for CFMPY-15. Enhances detail around methyl/methoxy substitutions.
(PDF)

**S14 Fig. Secondary NMR region for CFMPY-17.** Confirms consistency of multiple methoxy group signals.
(PDF)

**S15 Fig. Secondary NMR region for CFMPY-28.** Distinguishes ring protons from side-chain environments.
(PDF)

**S16 Fig. Chemical structures of pyrimidine-substituted chalcones.** Reference figure matching spectra with corresponding compound names and positions.
(PDF)

**S17 Fig. Raw uncropped gel image for DNA fragmentation assay (Fig 5).** Original gel documentation showing DNA fragmentation patterns in HeLa cells treated with synthesized compounds. Lanes: 1 – DNA Ladder (molecular weight marker), 2 - CFMPY-2, 3 - CFMPY-4, 4 - CFMPY-15, 5 - CFMPY-17, 6 - CFMPY-28, 7 – Cisplatin (positive control). This uncropped image is provided for transparency and to demonstrate the absence of gel manipulati.
(JPG)

## Author contributions

**Conceptualization:** Sunil Tulshiram Hajare.

**Data curation:** Prashant Nayak, Vikram S. Shenoy, Vijay Upadhye, Kasim Sakran Abass, Mukesh Soni.

**Formal analysis:** Prashant Nayak, Vikram S. Shenoy, Vijay Upadhye, Kasim Sakran Abass.

**Investigation:** Prashant Nayak.

**Methodology:** Sunil Tulshiram Hajare.

**Resources:** Kasim Sakran Abass, Sunil Tulshiram Hajare.

**Software:** Prashant Nayak, Vikram S. Shenoy, Vijay Upadhye, Kasim Sakran Abass, Mukesh Soni.

**Supervision:** Sunil Tulshiram Hajare.

**Validation:** Prashant Nayak, Vikram S. Shenoy, Vijay Upadhye, Kasim Sakran Abass, Omar Awad Alsaidan, Mukesh Soni, Sunil Tulshiram Hajare.

**Visualization:** Prashant Nayak, Vikram S. Shenoy, Vijay Upadhye, Kasim Sakran Abass, Omar Awad Alsaidan, Mukesh Soni, Sunil Tulshiram Hajare.

**Writing – original draft:** Sunil Tulshiram Hajare.

**Writing – review & editing:** Omar Awad Alsaidan, Sunil Tulshiram Hajare.

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
