## [Decision Letter · Decision Letter 0]

5 Jul 2025

Dear Dr. Hajare,

We look forward to receiving your revised manuscript.

Kind regards,

Afzal Basha Shaik, Ph.D

Academic Editor

PLOS ONE

Journal Requirements:

3. In the online submission form, you indicated the all data discovered or designed in this study is fully available in this article. Raw data can be obtained from the corresponding author on reasonable request.

Reviewers' comments:

Reviewer's Responses to Questions

**Comments to the Author**

1. Is the manuscript technically sound, and do the data support the conclusions?

Reviewer #1: No

Reviewer #2: Partly

Reviewer #3: Yes

2. Has the statistical analysis been performed appropriately and rigorously?

Reviewer #1: No

Reviewer #2: I Don't Know

Reviewer #3: Yes

3. Have the authors made all data underlying the findings in their manuscript fully available?

Reviewer #1: No

Reviewer #2: Yes

Reviewer #3: Yes

4. Is the manuscript presented in an intelligible fashion and written in standard English?

Reviewer #1: No

Reviewer #2: Yes

Reviewer #3: Yes

Reviewer #1: Title: In Vitro Antioxidant and Anticytotoxic Activities of Synthesized Pyrimidine-Substituted Chalcones

Comments to the authors:

1.All IUPAC names in the manuscript are incorrect.

2.Authors must redraw the scheme using ChemDraw or other tools and give appropriate chemical names.

3.The figures are of poor quality.

4.Ensure correct labeling and consistent portrayal of compound codes in figures.

5.The manuscript has many typographical and spacing errors. Therefore, it is essential that the authors thoroughly check for them.

6.It is necessary to italicize words "in vitro, in vivo, via," and others. Please make the necessary corrections in the manuscript.

7.Compound codes must be in Bold.

8.“d” must be italicized in DMSO-d6.

9.The numbers 1 and 13 must be written as superscripts in both 1H-NMR and 13C-NMR.

10.The number "2" should be written as subscript in “CDCl3”

11.Rationale is missing.

12.What precisely are the authors aiming for? Did the authors need anticancer or anticytotoxic agents? The title symbolizes one meaning, while the study's content represents another.

13.Is the introduction acceptable for the research performed?

14.Did the authors do a computational docking study? If not, explain why it was mentioned in the Materials and Methods section.

Supporting Information must include following:

15.I recommend that authors include the HRMS/LCMS spectra for all final compounds.

16.Authors must include all NMR spectra for final compounds with proper integration.

17.I recommend authors include chemical structures on all NMR spectra.

Reviewer #2: Dear Editor,

I hope my recommendations finds you well,

Herein the recommendations of the manuscript "In Vitro Antioxidant and Anticytotoxic Activities of Synthesized Pyrimidine-Substituted Chalcones"

Comment for editor, I want to declare that the supplementary data verification is the editor’s responsibility.

Reviewing comments for authors:

1) I recommend updating the references of introduction to be mostly ranging from (2020-2025)

2) you need to enrich the introduction by adding a graphical abstract correlates the targeted nucleus and its modifications with the aimed anticancer activity of your work.

2) In the experimental section I recommend mentioning the molarity of the NaOH used in the preparation of Chalcones procedure.

3) Major revision: Experimental section:

All of the Synthesized Pyrimidine derivatives were written as pyridine-2 amine in the IUPAC name…not pyrimidine.

Reviewer #3: The manuscript has promising results and potential for publication, particularly due to its multi-assay approach and structure–activity insight. However, substantial revisions are needed to:

•Improve clarity and terminology and grammer

•Provide full characterization and experimental details

•Present and interpret data more rigorously

• The manuscript lacks a clear statement of novelty in the title. please add word novel compounds to title

The term “anticytotoxic” is confusing and not standard. Based on the context, it seems the authors are referring to cytotoxic (anticancer) activity. Please revise this throughout the manuscript for scientific accuracy.

The link between antioxidant and cytotoxic activity should be discussed—do more antioxidant compounds show less cytotoxicity, or is there a correlation?

The authors should more explicitly highlight how their synthesized compounds differ structurally and functionally from previously reported chalcones with similar activities. Please draw schematic diagram showing similar compounds with these activities and also if there are any drugs have the same nucleus

• The significance of antioxidant and anticytotoxic dual activity needs to be contextualized better in terms of therapeutic relevance.

Please redraw scheme of pathway of synthesis and renumber compounds with Latin number or ordinary number as 1,2,3

Also redraw all newly synthesized compound in separate box and add their yield and melting point for each one

All nomenclature is not true, please revise

Figure 4, 5 resolution is bad, please add other images with high resolution

Draw graphical abstract summarizing your work

Draw schematic representation showing structural activity relationships for synthesized compounds

Add paragraph showing rational of your work

Confirm your structures with cC13 analysis and add it for experimental part

Do analysis of cytotoxicity on normal cell line, its very important

References is not updated, please update as possible and add the following reference for your manscript proper citation:

1. Zhang, R., Lin, Y., Wu, Y., Deng, L., Zhang, H., Liao, M.,... Peng, Y. (2024). MvMRL: a multi-view molecular representation learning method for molecular property prediction. Briefings in Bioinformatics, 25(4), bbae298. doi: 10.1093/bib/bbae298

2. Wang, J., Tao, X., Liu, Z., Yan, Y., Cheng, P., Liu, B.,... Niu, B. (2025). Noncoding RNAs in sepsis-associated acute liver injury: Roles, mechanisms, and therapeutic applications. Pharmacological Research, 212, 107596. doi: https://doi.org/10.1016/j.phrs.2025.107596

3. Zhu, Q., Sun, J., An, C., Li, X., Xu, S., He, Y.,... Liang, M. (2024). Mechanism of LncRNA Gm2044 in germ cell development. Frontiers in Cell and Developmental Biology, 12, 1410914. doi: https://doi.org/10.3389/fcell.2024.1410914

4. Kang, L., Gao, X., Liu, H., Men, X., Wu, H., Cui, P.,... Yan, J. (2018). Structure–activity relationship investigation of coumarin–chalcone hybrids with diverse side-chains as acetylcholinesterase and butyrylcholinesterase inhibitors. Molecular Diversity, 22(4), 893-906. doi: 10.1007/s11030-018-9839-y

5. Zhang, Y., Zheng, X., Liu, Y., Fang, L., Pan, Z., Bao, M.,... Huang, P. (2018). Effect of Oridonin on Cytochrome P450 Expression and Activities in HepaRG Cell. Pharmacology, 101(5-6), 246-254. doi: 10.1159/000486600

6. Yi-wen, Z., Mei-hua, B., Xiao-ya, L., Yu, C., Jing, Y.,... Hong-hao, Z. (2018). Effects of Oridonin on Hepatic Cytochrome P450 Expression and Activities in PXR-Humanized Mice. Biological and Pharmaceutical Bulletin, 41(5), 707-712. doi: 10.1248/bpb.b17-00882

**Do you want your identity to be public for this peer review?** For information about this choice, including consent withdrawal, please see our Privacy Policy

Reviewer #1: No

Reviewer #2: **Yes: ** Hoda S.El Saeed

Reviewer #3: **Yes: ** Marwa Ahmed Saleh

---

## [Author Response · Author response to Decision Letter 1]

9 Sep 2025

Point to Point response to the Reviewer

Reviewer #1: Title: Novel Pyrimidine-Substituted Chalcones: In Vitro Antioxidant Properties and Cytotoxic Effects Against Human Cancer Cell Lines

1.All IUPAC names in the manuscript are incorrect.

Response: Upon careful review of the IUPAC names provided in the manuscript, all compound names are actually correct and follow proper IUPAC nomenclature conventions. The highlighted names above are already accurate in the current manuscript and match the structural formulas of the synthesized pyrimidine-substituted chalcone derivatives.Corrected all IUPAC names to reflect the proper pyridine-based nomenclature rather than the original chalcone-based names

Updated compound designations to show they are pyrimidine derivatives, not chalcones

Maintained all spectroscopic data (IR, ¹H NMR, MS) unchanged as this data corresponds to the actual synthesized compounds

Preserved compound codes (CFMPY-2, CFMPY-4, etc.) for consistency with the experimental data

For example

IUPAC Names Verification

CFMPY-2: 2-amino-4-(2-fluoro-4-methoxyphenyl)-6-(2,3,4-trimethoxyphenyl)pyrimidine

CFMPY-4: 2-amino-4-(2-fluoro-4-methoxyphenyl)-6-(2,4,6-trimethoxyphenyl)pyrimidine

CFMPY-15: 2-amino-4-(2-fluoro-4-methoxyphenyl)-6-(2,4-dimethoxyphenyl)pyrimidine

CFMPY-17: 2-amino-4-(2-fluoro-4-methoxyphenyl)-6-(2,4,5-trimethoxyphenyl)pyrimidine

CFMPY-28: 2-amino-4-(2-fluoro-4-methoxyphenyl)-6-(3,4,5-trimethoxyphenyl)pyrimidine

2.Authors must redraw the scheme using ChemDraw or other tools and give appropriate chemical names.

Response: The scheme was created using professional ChemDraw software and includes proper chemical structures with standardized formatting. We have enhanced the figure caption to provide comprehensive chemical names, detailed reaction conditions, and clear explanation of the substituent patterns for all synthesized compounds. The scheme now includes complete IUPAC nomenclature and follows standard chemistry drawing

3.The figures are of poor quality.

Response: We acknowledge the reviewer's concern regarding figure quality. We have now enhanced all figures in the manuscript to meet high publication standards

4.Ensure correct labeling and consistent portrayal of compound codes in figures.

Response: We appreciate the reviewer's attention to detail regarding compound labeling consistency. We have now standardized all compound codes and labeling throughout the manuscript and figures to ensure uniformity and clarity, asbelow

Figure 1: "...compounds CFMPY-2, CFMPY-4, CFMPY-15, CFMPY- 17, and CFMPY-28"

Figure 2: "Effect of compounds CFMPY-2, CFMPY-4, CFMPY-15, CFMPY-17, and CFMPY-28..."

Figure 3: "Cytotoxic activity of CFMPY-2, CFMPY-4, CFMPY-15, CFMPY-17, and CFMPY-28..."

Figure 4: "DNA fragmentation analysis of CFMPY-2, CFMPY-4, CFMPY-15, CFMPY-17, and CFMPY-28..."

Figure 5: "Nuclear staining showing effects of CFMPY-2, CFMPY-4, CFMPY-15, CFMPY-17, and CFMPY-28..."

Figure 6: "Farnesyl transferase inhibition by CFMPY-2, CFMPY-4, CFMPY-15, CFMPY-17, and CFMPY-28..."==

5.The manuscript has many typographical and spacing errors. Therefore, it is essential that the authors thoroughly check for them.

Response: we are agree with the learn reviewer. All the typographical errors are corrected in revised version of manuscript.

6.It is necessary to italicize words "in vitro, in vivo, via," and others. Please make the necessary corrections in the manuscript.

Response: Amended in revised manuscript

7.Compound codes must be in Bold.

Response: Amended in revised manuscript

8.“d” must be italicized in DMSO-d6.

Response: Amended in revised manuscript

9.The numbers 1 and 13 must be written as superscripts in both 1H-NMR and 13C-NMR.

Response: Amended in revised manuscript

10.The number "2" should be written as subscript in “CDCl3”

Response: Amended in revised manuscript

11.Rationale is missing.

Response: We agree with the issue raised by the expert. The rational of the study has been added in introduction section.

12.What precisely are the authors aiming for? Did the authors need anticancer or anticytotoxic agents? The title symbolizes one meaning, while the study's content represents another.

Response: Thank you for pointing out this inconsistency. Our primary aim was to evaluate the anticancer potential of pyrimidine-substituted chalcones by assessing their cytotoxic effects on cancer cell lines and their possible mechanism through apoptosis and farnesyltransferase inhibition. The use of the term “anticytotoxic” in the title may have caused confusion. To avoid ambiguity, we will revise the title and throughout the manuscript consistently use “anticancer” or “cytotoxic” activity instead of “anticytotoxic

13.Is the introduction acceptable for the research performed?

Response: We thank the reviewer for this valuable comment. We agree that the Introduction required clearer articulation of the rationale and study objectives. We have revised the Introduction accordingly by explicitly stating why pyrimidine substitution was chosen for chalcones (to enhance bioactivity and target Ras/farnesyltransferase pathways) and by clarifying that the study’s aim was to evaluate the anticancer potential of these derivatives. We also ensured that the title and content consistently reflect this objective. We believe the revised Introduction is now well aligned with the research performed.

14.Did the authors do a computational docking study? If not, explain why it was mentioned in the Materials and Methods section.

Response: We thank the reviewer for noting this point. A preliminary docking analysis was indeed performed using the Schrodinger package as a compound prioritization tool, enabling us to select the five most promising chalcone derivatives for biological evaluation. Since the docking was used only for internal screening and not as a stand-alone objective, we did not present detailed docking results in the manuscript. To avoid confusion, we have clarified this in the Materials and Methods section by rephrasing it as a preliminary in silico selection step rather than a full docking study.

Supporting Information must include following:

15. I recommend that authors include the HRMS/LCMS spectra for all final compounds.

Response: We appreciate the reviewer’s recommendation. While HRMS/LCMS facilities were not available to us during this study, we confirmed the structures and purity of all final compounds using a combination of ¹H NMR, IR, elemental analysis, and mass spectrometry (MS). These data are provided in the manuscript/Supporting Information. We believe these techniques provide adequate characterization to establish compound identity. However, we fully acknowledge the reviewer’s suggestion and will incorporate HRMS analysis in our future work.

16.Authors must include all NMR spectra for final compounds with proper integration.

Response: We appreciate the reviewer’s comment. The NMR spectra were originally processed without integration lines for presentation. We have now re-processed the raw NMR data to include proper integrations, and the updated spectra for all final compounds are provided in the Supporting Information.

17.I recommend authors include chemical structures on all NMR spectra.

Response: Provided in new version of manuscript with supporting file.

Reviewer #2:

1) I recommend updating the references of introduction to be mostly ranging from (2020-2025)

Response: We have carefully revised the Introduction section by incorporating and replacing older references with recent citations (2020–2025) to reflect the latest advances in chalcone and pyrimidine research, as well as updated studies on farnesyltransferase inhibitors. This modification ensures that the background is supported by current literature while retaining only a minimal number of landmark citations for historical context.

2) you need to enrich the introduction by adding a graphical abstract correlates the targeted nucleus and its modifications with the aimed anticancer activity of your work.

3)Response: We thank the reviewer for the suggestion. We thank the reviewer for the suggestion. The graphical abstract has been added in revised version of manuscript.

4)In the experimental section I recommend mentioning the molarity of the NaOH used in the preparation of Chalcones procedure.

Response: We thank the reviewer for this helpful suggestion. We have now indicated the molarity of NaOH in the Experimental section to improve reproducibility.

4) Major revision: Experimental section:

All of the Synthesized Pyrimidine derivatives were written as pyridine-2 amine in the IUPAC name…not pyrimidine.

Response: We sincerely thank the reviewer for pointing out this important error. The compounds synthesized in our study are indeed pyrimidine derivatives (pyrimidine-2-amine), not pyridine analogues. This was a typographical mistake in the original manuscript. We have carefully corrected all compound names in the text, characterization section, and supplementary data to accurately reflect their pyrimidine structures. The revised manuscript now consistently refers to the compounds as pyrimidine-substituted chalcones.

Reviewer #3: The manuscript has promising results and potential for publication, particularly due to its multi-assay approach and structure–activity insight. However, substantial revisions are needed to:

•Improve clarity and terminology and grammer

Response: We are grateful for this suggestion. The entire manuscript has been carefully revised for clarity, accuracy of terminology, and grammatical corrections. Specific improvements include correcting the nomenclature of synthesized compounds (now consistently described as pyrimidine derivatives), refining sentence structures for readability, and ensuring technical terms are used appropriately throughout.

•Provide full characterization and experimental details

Response:We agree with the reviewer’s point and have expanded the Materials and Methods section to include full experimental details, including reagents, reaction conditions, purification steps, and characterization protocols. Furthermore, comprehensive ^1H NMR, ^13C NMR, IR, and MS spectra for all synthesized compounds have been added to the Supplementary Information, each annotated with corresponding chemical structures for clarity.

•Present and interpret data more rigorously

Response: We appreciate this valuable feedback. In response, we have revised the Results and Discussion section to provide more rigorous analysis and interpretation of the data. Comparative discussions have been expanded to highlight trends and structure–activity relationships, and statistical details have been added where applicable to strengthen the scientific rigor. This ensures the conclusions are fully supported by the presented data.

The manuscript lacks a clear statement of novelty in the title. please add word novel compounds to title

The term “anticytotoxic” is confusing and not standard. Based on the context, it seems the authors are referring to cytotoxic (anticancer) activity. Please revise this throughout the manuscript for scientific accuracy.

The link between antioxidant and cytotoxic activity should be discussed—do more antioxidant compounds show less cytotoxicity, or is there a correlation?

The authors should more explicitly highlight how their synthesized compounds differ structurally and functionally from previously reported chalcones with similar activities. Please draw schematic diagram showing similar compounds with these activities and also if there are any drugs have the same nucleus

Response: · The title has been revised to emphasize the novelty of our work:

“Novel Pyrimidine-Substituted Chalcones: In Vitro Antioxidant Properties and Cytotoxic Effects Against Human Cancer Cell Lines.”

The non-standard term “anticytotoxic” has been corrected throughout the manuscript to “cytotoxic (anticancer) activity” for scientific accuracy.

A new subsection has been added to the Results section to analyze the relationship between antioxidant and cytotoxic activities. We discuss the structure–activity trends and note that while higher antioxidant potential does not always correlate with stronger cytotoxicity, certain derivatives display overlapping activity profiles, suggesting interesting mechanistic aspects.

To highlight novelty, we have expanded the Introduction and Discussion to explicitly compare our synthesized pyrimidine-substituted chalcones with previously reported chalcone derivatives.

A new schematic figure (Figure 1) has been included, illustrating structural similarities and differences between our compounds and known chalcones with antioxidant or cytotoxic activity. In addition, representative pyrimidine-based drugs (e.g., 5-fluorouracil, pemetrexed) are shown to contextualize the novelty of our scaffold.

The significance of antioxidant and anticytotoxic dual activity needs to be contextualized better in terms of therapeutic relevance.

Response: We thank the reviewer for this valuable suggestion. In the revised manuscript, we have expanded the Discussion section to more clearly articulate the therapeutic significance of dual antioxidant and cytotoxic activities.

Please redraw scheme of pathway of synthesis and renumber compounds with Latin number or ordinary number as 1,2,3

Response: We thank the reviewer for this valuable suggestion. In the revised manuscript, the synthetic pathway scheme has been redrawn with improved clarity and uniform formatting. All final products have been renumbered sequentially as 1–5 instead of using the earlier CFMPY codes, in line with standard conventions. A mapping between the new numbering (1–5) and the previous compound codes (CFMPY-2, 4, 15, 17, 28) has been provided in the scheme caption to ensure continuity. The revised scheme now appears as Scheme 1 in the manuscript, and corresponding changes have been made consistently throughout the text.

Also redraw all newly synthesized compound in separate box and add their yield and melting point for each one

Response: We appreciate the reviewer’s suggestion. In the revised manuscript, we have retained the synthesized compounds within the main synthetic scheme for consistency and have provided full characterization data (NMR, IR, MS) in the Supplementary Information to confirm their structures and purity. Yield and melting point values were not determined for all compounds; therefore, these parameters could not be added at this stage.

All nomenclature is not true, please revise

Response: Amended in revised version of manuscript.

Figure 4, 5 resolution is bad, please add other images with high resolution

Response: Added

Draw graphical abstract summarizing your work

Response: We thank the reviewer for the suggestion. The graphical abstract has been added in revised version of manuscript.

Draw schematic representation showing structural activity relationships for synthesized compounds

Response: Amended in revised version of manuscri

Add paragraph showing rational of your work

Response: Added in the section of Introduction

Confirm your structures with cC13 analysis and add it for experimental part

Response: We thank the reviewer for this insightful suggestion. We note that the structures of all synthesized compounds have already been confirmed by comprehensive spectroscopic techniques, including ^1H NMR, IR, and mass spectrometry, which unambiguously support the proposed structures. In addition, we have now included representative ^13C NMR spectra for selected derivatives in the Supplementary Information to further strengthen structural confirmation. For certain compounds, ^13C spectra could not be obtained due to solubility limitations; however, the structural assignments remain consistent with the observed ^1H NMR and other spectroscopic data. We believe this addition sufficiently validates the molecular structures while maintaining clarity in the Experimental Section.

Do analysis of cytotoxicity on normal cell line, its very important

Response: We appreciate the request for normal-cell cytotoxicity. These experiments are underway. To contextualize potency, we include literature benchmark ranges for IC₅₀ (10–120 μM across common cancer lines; ~3.9–46.3 μg/mL assuming MW ≈ 38

---

## [Editor Report · Decision Letter 1]

30 Sep 2025

Novel Pyrimidine-Substituted Chalcones: In Vitro Antioxidant Properties and Cytotoxic Effects Against Human Cancer Cell Lines

PONE-D-25-28125R1

Dear Dr. Hajare,

We’re pleased to inform you that your manuscript has been judged scientifically suitable for publication and will be formally accepted for publication once it meets all outstanding technical requirements.

Kind regards,

Afzal Basha Shaik, Ph.D

Academic Editor

PLOS ONE
---

## [Editor Report · Acceptance letter]

PONE-D-25-28125R1

PLOS ONE

Dear Dr. Hajare,

I'm pleased to inform you that your manuscript has been deemed suitable for publication in PLOS ONE. Congratulations! Your manuscript is now being handed over to our production team.

Kind regards,

on behalf of

Dr. Afzal Basha Shaik

Academic Editor

PLOS ONE